# Developing Bioactive Hydrogels with Peptides for Dental Application

**DOI:** 10.3390/biomedicines12030694

**Published:** 2024-03-21

**Authors:** Alexandrina Muntean, Codruta Sarosi, Ioan Petean, Stanca Cuc, Rahela Carpa, Ioana Andreea Chis, Aranka Ilea, Ada Gabriela Delean, Marioara Moldovan

**Affiliations:** 1Department of Paediatric Dentistry, Iuliu Hatieganu University of Medicine and Pharmacy, 31 A. Iancu Street, 400083 Cluj-Napoca, Romania; alexandrina.muntean@umfcluj.ro (A.M.); chis.ioana.andreea@elearn.umfcluj.ro (I.A.C.); 2Department of Polymer Composites, Institute of Chemistry Raluca Ripan, Babes Bolyai University, 30 Fantanele Street, 400294 Cluj-Napoca, Romania; stanca.boboia@ubbcluj.ro (S.C.); marioara.moldovan@ubbcluj.ro (M.M.); 3Faculty of Chemistry and Chemical Engineering, Babes Bolyai University, 11 Arany Janos Street, 400028 Cluj-Napoca, Romania; ioan.petean@ubbcluj.ro; 4Department of Molecular Biology and Biotechnology, Faculty of Biology and Geology, Babes Bolyai University, 1 M. Kogalniceanu Street, 400084 Cluj-Napoca, Romania; rahela.carpa@ubbcluj.ro; 5Department of Oral Rehabilitation, Iuliu Hatieganu University of Medicine and Pharmacy, 15 Victor Babes Street, 400012 Cluj-Napoca, Romania; aranka.ilea@umfcluj.ro; 6Department of Cariology, Endodontics and Oral Pathology, Iuliu Hațieganu University of Medicine and Pharmacy, 33 Moților Street, 400001 Cluj-Napoca, Romania; ada.delean@umfcluj.ro

**Keywords:** peptides, hydrogels, hydroxyapatite, remineralisation, cytotoxicity

## Abstract

Dental caries is an avoidable and complex condition impacting billions of individuals worldwide, posing a specific concern among younger generations, despite the progress of oral hygiene products. This deterioration occurs due to the acid demineralization of tooth enamel, leading to the loss of minerals from the enamel subsurface. The remineralisation of early enamel carious lesions could prevent the cavitation of teeth. The enamel protein amelogenin constitutes 90% of the total enamel matrix protein and plays a key role in the bio mineralisation process. The aim of this study is to investigate the self-assembly microstructure and reticulation behaviour of a newly developed bioactive hydrogel with leucine-rich amelogenin peptide (LRAP) intended for enamel remineralisation. SEM, AFM, UV-VIS, and FTIR analyses emphasize the ability of peptides to promote cell adhesion and the treatment of early carious lesions. In conclusion, short-chain peptides can be used in hydrogels for individual or professional use.

## 1. Introduction

Dental caries represents a major public health issue. Enamel lesions are revealed as gradual subsurface demineralization that can lead to cavitation if not reversed [1,2,3]. A consensus states that the management of carious lesions should be “to preserve the tooth structure and restore only when necessary” [2,4].

In the oral cavity, demineralization and remineralisation processes frequently and regularly occur in a balanced way [5,6]. This balance depends on diet, hygiene habits, microbial activity, and on the overall general health status. All these factors can lead to enamel demineralization and the loss of tooth minerals, which favours the emergence of a cavity [6,7,8].

The non-intrusive, biomimetic early dental caries therapy is a promising and achievable goal, due to the recent advances in enamel remineralisation. Remineralisation systems can help regenerate the enamel microstructure, which can prevent the progression of decay and fully restore the strength and function of the tooth structures [1,2,3,9,10].

Current remineralisation systems are either based on the enamel amelogenin-like proteins (e.g., leucine-rich amelogenin peptides, amelogenin derivatives), dentine phosphoprotein (DPP), or on synthetic peptides that are similar to salivary proteins (e.g., P11-4) [1,10].

While fluoride remains the gold standard for caries management and enamel remineralisation with favourable clinical results, these newly developed systems are comparable to the action of fluoride, whilst reducing the potential risks associated with fluoride overdose [1,2,3].

The remineralisation of enamel and dentin using bioactive peptides has recently been considered as an alternative to the conventional treatment protocols using fluoride and amorphous calcium phosphate-based products.

Enhancing the knowledge of the kinetics of mineral loss and remineralisation can be accomplished by simulating dental diseases in vitro and in situ. Different dental lesion types and pH-cycling model systems have been developed to imitate and study decay progression [2]. To get the best results, these experiments must be performed in a sterile, contamination-free setting and run for a specific amount of time. Current remineralisation techniques seek to transform the therapeutic approach by restoring superficial enamel in line with minimally invasive dentistry outcomes [3,4]. A critical understanding of the operative role of naturally occurring enamel-forming proteins (such amelogenin) in triggering apatite mineralization in vitro is necessary to achieve this bioinspired approach [5,6,7,11,12,13,14,15]. These studies are meant to achieve a special material characterised by the hindrance of tooth decay, which is able to restore and replace the enamel that was lost, thus prolonging the integrity of the tooth.

Enamel remineralization involves the restoration of minerals such as hydroxyapatite to the tooth enamel, which is essential for preventing and repairing early stages of tooth decay. Peptides are short chains of amino acids that can play a role in enamel remineralization by promoting the attachment of the hydroxyapatite nano-crystals from the filler nanoparticles through the protein units. Thus, the connection between peptide and existing protein units within the demineralized enamel assures the proper growth of the newly formed hydroxyapatite nano-structural units. Hydrogels, on the other hand, provide a suitable delivery system for peptides, allowing for controlled release and prolonged contact with the tooth surface. The microstructural self-assembly of the gels in thin films plays an important role in remineralization success and requires detailed investigation. The aim of this study is to assess the self-assembly microstructure and reticulation behaviour of a newly developed bioactive hydrogel with leucine-rich amelogenin peptide (LRAP) intended for enamel remineralisation. It is expected that the peptide generates a network within the gel’s thin films applied onto a solid substrate, which is able to assure a uniform distribution of the mineral filler micro- and nanoparticles. This study is very important for figuring out the microstructural enhancement prior to testing on the human enamel surface. The general view of the microstructural aspects could be observed using SEM microscopy in optimal conditions but the fine microstructure and nanostructure of the gel’s self-assembly required AFM investigation. Peptides were highlighted in the experimental hydrogels through physio-chemical, antibacterial, and cytotoxicity tests.

## 2. Materials and Methods

### 2.1. Preparation of Hydrogels with Peptides

The peptides involved in this research were manufactured by Synpeptide Co., Ltd. (Nanjing, China) through conventional solid-phase peptide synthesis. Subsequently, they underwent purification and identification utilizing HPLC and ESI-MS. These peptides correspond to the amino acid sequences of porcine amelogenin P173. Peptide 1 (EMD Millipore Corp, Billerica, MA, USA Affiliate of Merck KGaA, Darmstadt, Germany) was synthesized from the N- and C-termini of porcine amelogenin, while peptide 2 (Sigma-Aldrich, INC, Saint Louis, MO, USA) is the peptide TRAP synthesized. The protein was initially dissolved in deionized water to create a peptide stock solution [16]. This peptide stock solution was stirred for 24 h on a stirrer at 4 °C and then stored in a refrigerator for a minimum of 24 h.

The hydrogels were prepared from a mixture of Polyethylene glycol 400, PEG 400 (Sigma-Aldrich Inc., Darmstadt, Germany): fumed silica nanoparticles (Remed Prodimpex SRL, Bucharest, Romania) in a weight ratio 3.4:1, distilled water and hydroxyapatite (HA—synthetised in our laboratory). In the homogenous mixture 0.015% 2-Hydroxyethyl salicylate (Merck Schuchardt oHG, Hohenbrunn, Germany) was added.

The obtained hydrogel was divided into three equal parts, in which peptides Anti-Amelogenin, X isoform, ABT260, denoted as P1 (EMD Millipore Corp, Billerica, MA, USA Affiliate of Merck KGaA, Darmstadt, Germany), and Anti-Kallikrein L1, K3014, denoted as P2 (Sigma-Aldrich, INC, Saint Louis, MO, USA) were added. One part was maintained as control gel G0. The experimental hydrogel G3 was made using the same components with an addition of nano capsules with P1 peptide (Table 1).

Nano capsules were prepared by mixing two phases: organic and aqueous. For the organic phase, polylactic acid (PLA) (NatureWorks LLC, Plymouth, MN, USA under the Ingeo^®^ brand), Miglyol^®^ 810 N (IOI Oleo GmbH, Witten, Germany), acetonitrile (Sigma-Aldrich Inc.), 1% tricalcium phosphate with chitosan (synthetized in our laboratory), and 0.5% peptide 1 (Anti-Amelogenin, X isoform) were used. The aqueous solution consisted of Poloxamer 407 (Sigma-Aldrich Inc.), distilled water, and acetonitrile (Sigma-Aldrich Chemie GmbH, Taufkirchen, Germany). We added chitosan as a component in controlled release systems for peptides, helping maintain stability and achieve the gradual release of the active substance, namely, peptide.

### 2.2. FTIR Spectroscopy

The hydrogels were investigated through Fourier-transform infrared spectroscopy (FTIR) with a FTIR 610 spectrometer from Jasco Corporation in Tokyo, Japan. The analysis spanned the wavenumber range of 4000–400 cm^−1^ and employed the ATR technique. Spectra were obtained with a resolution of 4 cm^−1^, and 100 scans were repeated for each measurement. All FTIR spectra were recorded at room temperature, with a minimum of three determinations conducted for each sample.

### 2.3. UV-Vis Spectroscopy

For the investigation of peptides hydrogels, we used the double beam Jasco V-750 UV-Vis Spectrophotometer (JASCO Corporation, Tokyo, Japan) to determine the absorption bands of the hydrogel’s components. The measurements were conducted within the spectral range of 400 to 1000 nm, covering the visible and near-infrared domains. The measurements of reflectance on solid samples were made using a 150 mm integrating sphere, with double beam, on a support having a diameter of 150 mm and a film thickness of 0.1 mm. To assess the reproducibility of the measurements, all experiments were replicated three times.

### 2.4. Antibacterial Test

The antibacterial test was performed using Nutrient agar (Merck Life Science S.L.U., Madrid, Spain) and Mueller Hinton agar culture media (Merck Life Science S.L.U., Madrid, Spain), sterile saline, sterile Petri dishes, sterile micropipettes and tips, sterile tweezers, a Precisa analytical balance, a Consort pH meter, a Laminar flow hood with vertical air model BIO-M SCS 1-4, a Raypa electric autoclave, and a Salvisincucentric IC400 incubator.

The microorganisms tested in this study were *Streptococcus mutans* ATCC 25175, *Streptococcus thermophilus* ATCC 19258, and *Streptococcus salivarius* ATCC 13419 from the collection of the Microbiology Laboratory, Faculty of Biology and Geology, UBB, Cluj.

All these strains were cultivated for a duration of 24 h, on a medium with Nutrient Agar (Merck Life Science S.L.U., Madrid, Spain). Next, each strain was diluted to a concentration of 0.5 MCFarland in sterile physiological serum. A sterile brush soaked in the 0.5 McFarland microbial solution was used to inoculate each Petri dish having these dilutions as source, covering the whole surface of the culture medium (Mueller–Hinton Oxoid). Then, they were dehydrated for 15 min, at 37 °C.

From the samples of the prepared test gels (G1, G2, and G3) and the control gel (G0) an amount of approximately 5 mm in diameter was placed with a micro spatula.

### 2.5. Cytotoxicity Test

The cytotoxicity evaluation was assessed using co-culture techniques. The hydrogels were added onto the cell culture inserts (Falcon^®^) and cells were added to the culture surface of 12-well plates. The HFL 1 (CCL-153™) cells were seeded at a concentration of 1 × 10^5^ cells in complete propagation medium DMEM/F12 (Gibco Life Technologies, Paisley, UK) supplemented with 10% foetal bovine serum (FBS) (Hyclone, Fisher Scientific, Leicester, UK) and 1% antibiotic-antimycotic (Gibco Life Technologies, Paisley, UK). After 24 h, the selected hydrogels were added onto the inserts and incubated for 24 h. Cells maintained in the normal propagation medium were used as control. Each experimental condition was replicated three times. After 24 h, the culture medium and the inserts were removed and 100 µL of MTT solution (1 mg/mL) (Sigma-Aldrich, St. Louis, MO, USA) was added. Following a 4 h incubation at 37 °C, the MTT solution was extracted, and 100 µL of DMSO (dimethyl sulfoxide) solution (Fluka, Buchs, Switzerland) was introduced. The optical density of chromogenic reactions was evaluated using the BioTek Synergy 2 spectrophotometer (Winooski, VT, USA) at a 450 nm wavelength.

The calculation of the results was carried out by reporting the optical density compared to the control. The one-way ANOVA test was employed to assess if there were statistically significant differences among the groups.

### 2.6. SEM and AFM Microscopy

The investigated hydrogels were applied on the solid substrate (E.G. optical glass slides) as a drop, which was spread with a spatula into a thin and uniform film, and analysed through Scanning Electron Microscopy (SEM) and Atomic Force Microscopy (AFM).

SEM investigation was performed using an Inspect™ SEM microscope produced by FEI Company, Hillsboro, OR, USA, operated in the low vacuum mode at an acceleration voltage of 20 kV. Secondary electron (SE) images were obtained at high resolution at different magnifications to reveal the overall aspect of the samples and their fine microstructure.

Atomic Force Microscopy (AFM) was carried out on a JEOL JSPM 4210 Scanning Probe Microscope, produced by JEOL, Tokyo, Japan. Samples were examined using NSC 15 cantilevers produced by MikroMasch, Tallinn, Estonia. They exhibit a resonant frequency of 325 kHz and a force constant of 40 N/m. The topographic images were acquired over an area of 20 μm × 20 μm to better observe the fine microstructural aspects previously visualized using SEM and at 5 μm × 5 μm for the nanostructural aspects observation. The scan rate was situated at about 1.5 to 3 Hz depending on the scanned area and local sample characteristics. The specialized software Jeol WIN SPM 2.0 produced by JEOL, Tokyo, Japan, was used for AFM images analysis and for specific measurement. Therefore, the surface roughness parameters Ra and Rq were calculated using the specific equations previously described in the literature [1,2].

At least three distinct macroscopic areas were scanned for each sample to ensure a reliable statistical average of the obtained values.

## 3. Results

### 3.1. FTIR Spectroscopy

The obtained hydrogels were investigated through FTIR spectroscopy to provide data for the secondary structure characterization of the peptides.

The FTIR spectra of the peptides, control, and hydrogel with peptides are shown in Figure 1a, b. In the control (G0), the FTIR spectrum of the hydrogel showed bands around 3367 cm^−1^, 1645 cm^−1^, 1066 cm^−1^, and 673 cm^−1^. Silica-based hydrogels showed intense absorption peaks around 3711, 3032 cm^−1^, and 1645 cm^−1^ values. These maxima of absorption around the values of 3367 cm^−1^ and 1647 cm^−1^ are due to the vibrations of the O-H bonds in water.

The characteristic bands of PEG depend on the degree of polymerization, and can be located according to [17] at 3441 cm^−1^ in the case of OH stretching vibrations, 2878 cm^−1^ for C-H stretching, 1464 cm^−1^ and 1343 cm^−1^ for CH deformation, and 1094 cm^−1^ for O-H and C-O-H. It seems that the bands, due to the presence of peptides, play a less important role. There is probably a destruction of the three-dimensional structure due to the appearance of stronger hydrogen bonds of the C=O···O-H type, which are formed between polyethylene glycol and peptides, replacing the C=O···N-H bonds in the peptides, and interactions of the peptides with the silica are also possible.

The FTIR spectra of the peptides (P1 and P2) and hydrogels with peptides (P1 and P2) closely resembled the FTIR spectrum (1650 cm^−1^ and 1450 cm^−1^) presented by Warren et al. of bovine pancreatic trypsin inhibitor, representing the Amide I and Amide II bands [18].

The FTIR-ATR spectra highlight the presence of peptides in the experimental gels at the specific bands at Amide I (1645 cm^−1^), C=O stretching, Amide A (3367 cm^−1^). The absorption associated with the Amide I band leads to stretching vibrations of the C=O bond of the amide, while the absorption associated with the Amide II band leads primarily to bending vibrations of the N-H bond.

In the G0, G1, and G2 spectra, absorption bands at approximately 960 cm^−1^ are evidenced and can be associated with vibrations of the PO_4_^3−^ group from the hydroxyapatite. The peaks at 570 and 601 cm^−1^ correspond to stretching vibrations of phosphate and the peak. The absorption band at approximately 630–650 cm^−1^ is associated with vibrational deformations of the PO_4_^3−^ groups in the crystalline lattice of hydroxyapatite. The presence of the peak at 1462 cm^−1^ is related to CO_3_^2−^ group and suggests that carbon from the organics does not pyrolyse completely and may instead dissolve into a HA crystal.

### 3.2. UV-Vis Spectroscopy

UV absorption spectroscopy is a widely utilized method for peptides to assess concentration and enzyme activity. Nevertheless, high-resolution UV spectra can also yield insights into the secondary and tertiary structure of peptides as well as their association behaviour.

Recent advancements involving high-resolution second derivative absorption methods, which are temperature- and cation-dependent, can also furnish insights into peptide dynamics. Information from various low-resolution spectroscopic methods, including UV absorption, can be integrated to form a comprehensive view of peptide structure in response to varying environmental conditions.

In the experimental hydrogels, G1 and G3 containing the P1 peptide had an absorption peak at 219 nm (Figure 2). The intensity of the absorption peak at 219 nm increases for the hydrogels that contain P1 peptide, namely, for the G1 and G3 hydrogel, compared with G2 which do not have such an evident absorption band. The spectrum for the G0 hydrogel presents a low-intensity absorption band at 219 nm. In the spectra of G1, an absorption peak appears at 269 nm, and for the G3 spectra, the absorption peak at 259 nm is not so intense.

### 3.3. Antibacterial Test

After the incubation period at 37 °C was ended, the inhibition zones (mm) of the tested microbial strains were determined. It was observed that for all strains, the control sample (G0) did not show inhibition, but the tested samples differed in inhibition diameter size based on the tested amicrobial strain (Figure 3).

For *S. mutans* strains, the one-way ANOVA test shows significant differences between the gels, and the value of *p* = 1.0532 × 10^−5^.

As regards *Streptococcus mutans* ATCC 25175, inhibition was noticed in all samples, with a high result for G1 and G3 (gel with P1) (9 mm and 10 mm, respectively).

In sample G2 (gel with P2), a low inhibition was recorded. In the bacterial strain *Streptococcus salivarius* ATCC 13419, a good inhibition was observed in sample G3 (gel with P1) (10 mm). Further, at *Streptococcus thermophilus* ATCC 19258 strain, there was inhibition occurring in sample G3 (gel with nano capsules with the P1 peptide) (13 mm).

This was the highest value obtained for these samples. In samples G1 (gel with P1) and G2 (gel with P2), a low inhibition was recorded (Figure 3). The addition of nano capsules with the P1 peptide increased the antibacterial effect of hydrogel G3. Chitosan was used as a component in controlled release systems for peptides, helping maintain stability and achieve the gradual release of the active substance.

### 3.4. Cytotoxicity Test

After performing the one-way ANOVA test, no significant differences were observed among the gels. The results are presented in Table 2 and Table 3. The value of *p* = 0.12108.

### 3.5. SEM Microscopy

The SEM images obtained for the experimental hydrogels are presented in Figure 4.

The general aspect of the deposited thin films was observed. Hydrogel G0, containing only nanostructured hydroxyapatite (nHAP) and silica nSiO_2_, generates an island structure within the uniform thin film due to its spreading over the solid substrate (Figure 4a).

The formed islands have rounded-elongated shapes (some of them being ellipsoidal), with sizes varying from 15 to 30 µm, and their border presents some minor cracks.

The addition of peptide 1 in G1 sample causes the islands’ surface to increase their sizes, being situated over 100 µm, and the border cracks are more evident. The formed thin film compactness is increased as well as its uniformity. Several microstructural clusters are observed (Figure 4b). Peptide 2 within gel G2 is more effective in thin film spreading, generating large uniform domains with a compact deposition of the mineral filler particles (Figure 4c). These are organized in smaller clusters, which require further investigation at high magnification.

G3 contains small microcapsules uniformly spread on the thin film, having rounded shape and diameters ranging from 5 to 7 µm (Figure 4d). Most of them were fractured during the thin film deposition due to the tangential effort induced by the frictional forces. Thus, the microcapsules’ filler content was released into the gel matrix forming a uniform layer deposited on the solid substrate.

The thin films’ fine microstructure plays an important role in gel sample characterization. Figure 4e reveals the G0 fine microstructure which contains small micro clusters in the range of 1–3 µm which are locally agglomerated into bigger formations of about 5–7 µm. All these formations are uniformly spread within the gel matrix, assuring a good cohesion of the deposited film.

Peptide 1 acts as a local mediator within the small clusters, generating their slight increase to about 5 µm and the formation of local agglomerations up to 12 µm (Figure 4f). The high uniformity and compactness of the thin films mediated by peptide 2 leads to a very well-organised fine microstructure of the deposited thin film. The structural clusters are very fine, being almost indistinguishable, and their local agglomerations are situated in the range of 1–3 µm (Figure 4g).

The G3 samples’ fine microstructure reveals few microcapsules of about 7 µm: two of them have rounded shape and are positioned on the left side of the image, and a broken one is situated on the right side of the image (Figure 4h). There are also some small rounded microcapsules of about 3 µm, uniformly spread on the thin film surface. All these microcapsules are well embedded into a uniform and compact structure, with very fine constituents that require more enhanced microscopic techniques for their proper observation.

### 3.6. Atomic Force Microscopy (AFM)

Atomic Force Microscopy (AFM) gives a higher resolution of the fine microscopy insight. Thus, the topography of the G0 sample, Figure 5a, features a dense agglomeration of submicron clusters forming the microstructural features observed using SEM in Figure 4e. We can observe local heights of about 4.7 μm as seen in the 3D profile presented below the topographic image. Peptide 1’s presence in the G1 sample facilitates the fine clusters’ reticulation into the thin film, assuring a compact topography, with some excrescences of about 5 μm in diameter (as also observed using SEM) containing a dense agglomeration of submicron clusters (Figure 5b).

The reticulation effect is more enhanced by peptide 2’s action, as observed in Figure 5c. The filler clusters are very well spread among the gel matrix, having a submicron consistency assuring a uniform thin film with a regular topography, as observed in the corresponding 3D profile. Small microspheres within the G3 sample facilitate a uniform distribution of the TCP mineral filler, which is capped by chitosan forming a dense and compact layer embedding submicron clusters (Figure 5d). This is proof that chitosan’s presence improves the reticulation effect of peptide 1, which shows a better mediation of the filler particles’ distribution.

The samples’ nanostructure was observed at a scan size of 5 μm × 5 μm, allowing a better view of the mineral filler clusters. The control gel features large submicron clusters of about 800 nm strongly embedded into the gel matrix (Figure 5e). A relative uniform distribution of nano hydroxyapatite in the control gel is also observed that induces small nanostructured waves within the gel structure, with about 60 nm width and 180 nm length. This is related to the local segregation tendency in the control sample that facilitates the islands’ formation.

Peptide 1’s presence within the G1 sample considerably improves the nanostructure, as seen in Figure 5f. Better structured submicron clusters of about 600–800 nm is observed, embedded into the gel matrix. Free nanostructured hydroxyapatite is better reticulated into the thin film nanostructure based on a proactive effect of the peptide; thus, the waves’ structure is significantly reduced, with about 50 nm width and 100 nm length.

Peptide 2 has a more enhanced reticulation effect at the nanostructural level, as observed in Figure 5g. The submicron clusters are smaller, around 500 nm, and better embedded into the gel matrix, and the wavy formations related to the free nano HAP are significantly attenuated. Their presence was observed at the major borders of the cluster’s formation, and features a width around 40 nm and a length of about 80 nm.

The chitosan mediation of peptide 1 feature a nanostructural synergy during the embedding of small submicron clusters of the mineral filler released from the microcapsules. Figure 5h reveals rounded submicron clusters around 400 nm strongly embedded into the gel matrix. This is caused by the chitosan’s presence, which is well known for its ability to form dense and uniform thin films or foils [3,4]. Free nano HAP presence is not visible, being strongly embedded into the chitosan layer, a fact that improves gel stability after application on the desired surface.

Roughness characterizes the surface uniformity and the ability of the tested gels to form a smooth thin film. The fine microstructure level is affected by the island formation in the control gel, which has the higher roughness values, as observed in Figure 6a, which progressively decreases by the addition of peptide 1 and 2. It was noticed that peptide 2 generated lower roughness at the fine microstructure level. It was expected for the surface roughness to be lower in the case of G3 sample due to the chitosan and peptide 1 synergistic action, but it was slightly increased due to the remaining microcapsules present at the fine microstructural level.

The peptide reticulation of the submicron clusters facilitates the uniformities of thin film nanostructure, illustrated by the slow decrease in the roughness values induced by peptide 1 and the significant decrease in the roughness values induced by peptide 2 (Figure 6b). The three-dimensional reticulation of the G3 sample causes a significant increase in the roughness at the nanostructural level, but it does not affect the higher compaction of the structure.

## 4. Discussion

The purpose of this research was to create hydrogels with peptides for enamel remineralisation and to develop an efficient and innovative formula that facilitates the process of repairing and strengthening tooth enamel. The peptides included in the hydrogels are chosen for their specific properties, such as the ability to stimulate remineralisation and to enhance enamel resistance. This type of product can be used in dental treatments to indorse cavity prevention and to strengthen the enamel surface, providing significant benefits for oral health.

In the present research, the influence of the peptides from the experimental hydrogels through cell culture and microbial strains was observed. The presence of the peptides in the composition of the experimental hydrogels was confirmed using FTIR and UV-Vis spectroscopy. The surface properties of the hydrogels were investigated through SEM and AFM microscopy.

Absorption bands in the UV-Vis spectrum of a peptide can provide significant information about specific groups in its structure. In the FTIR spectra, specific functional groups of the compounds present in the composition of the hydrogels were identified. Peptides contain functional groups such as peptide bonds and amino acids that can absorb light at certain wavelengths. 

C=O···N-H bonds in peptides represent the hydrogen bonds formed between the carbonyl group (C=O) of one amino acid residue and the amino group (N-H) of another amino acid residue in a polypeptide chain. These hydrogen bonds are essential for protein structure and stability, helping to maintain a specific three-dimensional configuration of protein molecules. On the other hand, the small concentration of peptides in the hydrogels (0.5%) explains the lack of a clear absorption maximum [17].

The idea of choosing Amelogenin, X isoform, and Anti-Kallikrein L1 peptides was to improve the remineralisation of enamel and dentin.

The involvement of the Amelogenin, X isoform, in biomineralization is intensely studied. Specifically, it appears to manage the formation of crystallites during the secretory stage of tooth enamel development, playing a crucial role in both structural organization and mineralization. These peptides have the capacity to cause various cellular processes by impacting cell behaviour and altering growth properties. Consequently, this leads to the improvement of cell adhesion, proliferation, migration, differentiation, and biomineralization, which are essential considering the intricate nature of oral tissues [19]. Delivering therapeutic proteins and peptides orally establishes a challenge due to their low stability caused by the acidic pH.

Zhang et al. examined the interactions between calcium ions and peptides derived from amelogenin, suggesting that self-assembly may occur in the early stages of enamel development when calcium ions are present. The findings underscored the significance of calcium and phosphoserine interactions in influencing the patterns of protein–protein interactions [20].

Ding et al. investigated in vitro the remineralization of enamel caries through the combined use of an amelogenin-derived peptide and fluoride. The study assessed the effectiveness of this dual approach in promoting the restitution of demineralized enamel [7].

Mukherjee et al. suggested that the application of a hydrogel with chitosan and amelogenin-derived peptides can effectively promote enamel repair, demonstrating comparable efficiency to the full-length amelogenin–chitosan hydrogel [21,22,23,24].

In order to develop a clinically viable product for biomimetic enamel remineralization, we designed hydrogels containing amelogenin-derived peptides. Adding chitosan, we consider its protective biological and functional actions to ensure prolonged peptide viability in the oral cavity [21]. Chitosan is a product obtained through the alkaline deacetylation of chitin, a naturally occurring biopolymer derived from the exoskeleton of crustaceans, molluscs, insects, and certain fungi [25]. It is of commercial interest due to its biodegradability, biocompatibility, nontoxicity, antimicrobial activity, and adhesive properties [26]. Chitosan is used in the field of regenerative medicine and material research due to its versatile biological properties. In our study, we consider its role as a scaffold that could act as a template for crystal growth while presenting amelogenin proteins and peptides. Moreover, the limited solubility of chitosan at neutral pH affords a unique opportunity to deliver drugs, genes, and peptides [22].

Excessive sugar intake contributes to the creation of an acidic environment in dental cavities. This acidity, in turn, promotes bacterial colonization and causes a decrease in pH levels. In this environment, the demineralization process is initiated on the surface of the enamel [27]. To prevent these occurrences, several remineralising agents were taken into consideration [27]. An ideal agent should be non-toxic and capable of initiating remineralization without causing harm to the dental surface. Matrix-facilitated mineralization, resembling a natural process, is desired, yet most existing agents lack this capability [28]. The introduction of peptides has overcome this limitation, as they possess the ability to regenerate enamel. These agents initiate remineralization by constructing formations that mimic the extracellular matrix of the dental surface [19].

Considering their structure, peptides are consisting of a chain of amino acids linked together by peptide bonds. (AA) [29]. According to the literature, the sizes of peptides may vary from <20, <50, to <100 [12,19,27,28,29,30]. Peptides have garnered interest across different dental disciplines for their potential to induce biological effects in the oral environment [30].

Following enamel maturation, there is no ongoing production of enamel structural units, rendering the natural regeneration of damaged or destroyed tissue impossible. Due to the absence of healing through cellular repair mechanisms, the restoration of decayed enamel tissues in everyday practice relies on the physical and chemical process of remineralisation. Strategies involving bioglass, calcium phosphates, and fluoride have been utilized for the regeneration of enamel [12,30].

The process of enamel formation remains uncertain, despite numerous studies exploring the potential of amelogenin, the primary enamel matrix protein, in directing the growth of enamel crystals [31,32,33,34,35,36,37,38].

Despite the encouraging evidence for biomimetic remineralization in caries prevention through organic and inorganic interactions, there is a limited availability of products suitable for clinical applications. The properties exhibited by various peptide associations used in this study are in line with data in the literature and encourage the development of products capable of preserving and repairing enamel structure. Research indicates that complete recombinant amelogenins encompass crucial N- and C-terminal domains that are highly conserved and vital for enamel formation [36,37,38,39,40].

The leucine-rich amelogenin peptide (LRAP) is a variant of the full-length amelogenin protein, specifically an alternative splice variant consisting of 56 amino acid residues. LRAP retains the N- and C-terminal domains of the parent amelogenin but lacks the phosphorylated regions. LRAP has been identified for its involvement in mineralization processes, demonstrating the ability to induce mineralization akin to nonphosphorylated full-length amelogenins [41]. The application of LRAP in enamel remineralization has been investigated by Shafiei et al. [42], Mukherjee et al. [22], and Kwak et al. [43], using hydroxyapatite and chitosan as a biomimetic approach to the regeneration of human enamel. Our results are in line with this statement; both SEM and AFM analysis encourage the use of this peptide in products intended for oral hygiene. The synthetic peptides used in this study, P1 and P2, are intrinsically disordered and formed a distinctive nanostructured scaffold reminiscent of “nanospheres”, resembling the ones created by the complete amelogenin.

We conclude that the antimicrobial manifestation of peptide preparations against bacteria and the effect against *Streptococcus mutans* are encouraging results. Evidence in caries development places emphasis on the role of *S. mutans*; it possesses characteristics linked to dental caries and can attach to other oral bacteria as well as the acquired enamel pellicle (acidogenicity, aciduricity, and the ability to build exopolysaccharides from sucrose) [44,45,46].

In vitro results allow us to consider that peptide gels will help not only with enamel remineralisation but also microbial control, considering that glucidic intake is strongly related with patient behaviour.

While research in this field is ongoing, the combination of hydrogels and peptides shows promising results in developing effective strategies for enamel remineralization and addressing early stages of tooth decay. However, it is important to note that practical applications in clinical settings may require further research and validation. Limitation: As hydrogels can be functionalized with multiple bioactive molecules, and combined with other nanoparticles, another challenge will be determining the optimal ratio and compatibility of each component to produce synergetic effects.

Despite the number of studies on materials with peptides, more knowledge is needed regarding the effectiveness of this form of treatment. Studies that include new material targeted lesions, methodological standardization, reproducibility, and adequate data analysis are needed.

## 5. Conclusions

Examining the self-assembly microstructure and cross-linking behaviour, this study explored a newly developed bioactive leucine-rich amelogenic peptide (LRAP) hydrogel for enamel remineralization. This study involved preparing peptide hydrogel samples, creating peptide-controlled release capsules, and conducting structural, antibacterial, and cytotoxic characterizations. Notably, the antimicrobial effect of the peptide hydrogels against bacteria, particularly Streptococcus mutans, yielded promising results. In the cytotoxic test, no significant differences were observed among the hydrogels. This study being a pilot one on the novel peptide gel, further studies are needed to demonstrate remineralizing properties.

## Figures and Tables

**Figure 1 biomedicines-12-00694-f001:**
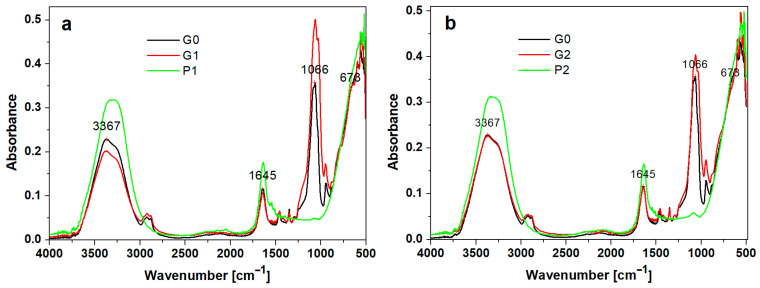
The FTIR spectra of investigated hydrogels: (**a**) black—G0 (control sample); red—G1 hydrogel with P1 peptide; green—P1 peptide; (**b**) black—G0 (control sample); red—G2 hydrogel with P2 peptide; green—P2 peptide.

**Figure 2 biomedicines-12-00694-f002:**
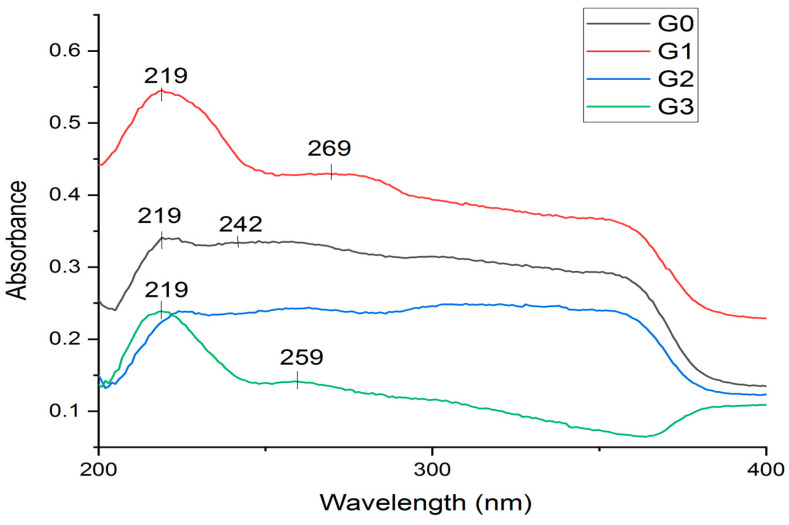
UV-Vis spectra of experimental hydrogels: control G0 (without peptide), and G1, G2, and G3 with P1 and P2 peptides.

**Figure 3 biomedicines-12-00694-f003:**
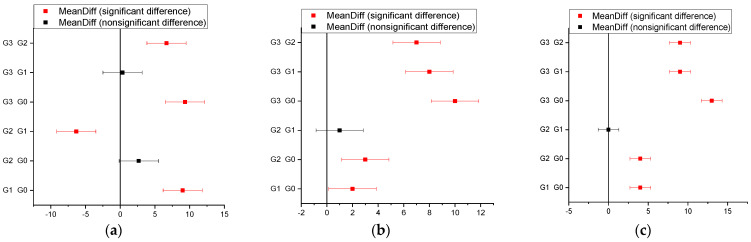
Tukey’s test of antibacterial activity of experimental hydrogels against *S.mutans* (**a**), *S.salivarius* (**b**), and *S.thermophillus* (**c**).

**Figure 4 biomedicines-12-00694-f004:**
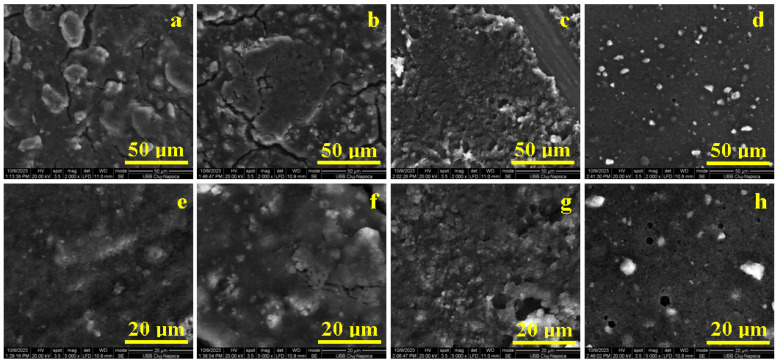
SEM images of the experimental hydrogels’ general aspect: (**a**) G0, (**b**) G1, (**c**) G2, and (**d**) G3; and fine microstructure: (**e**) G0, (**f**) G1, (**g**) G2, and (**h**) G3.

**Figure 5 biomedicines-12-00694-f005:**
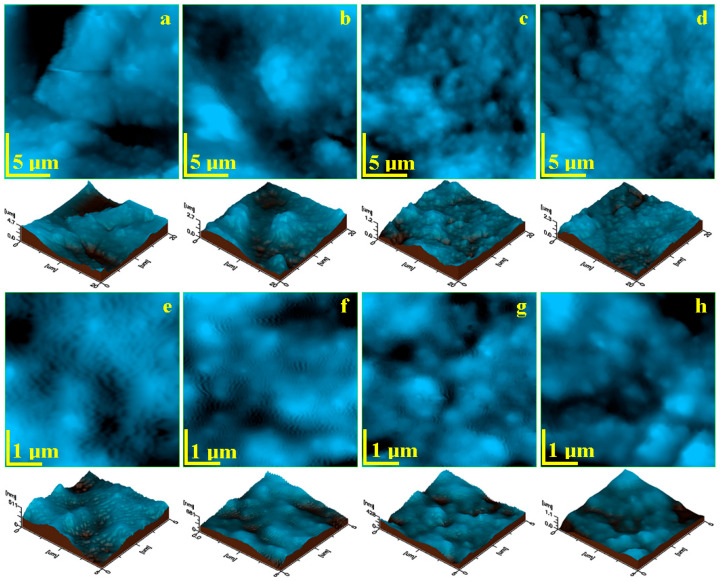
AFM topographic images of the experimental hydrogels’ fine microstructure: (**a**) G0, (**b**) G1, (**c**) G2, and (**d**) G3; and nanostructure: (**e**) G0, (**f**) G1, (**g**) G2, and (**h**) G3. Three-dimensional profiles are given below each topographic image.

**Figure 6 biomedicines-12-00694-f006:**
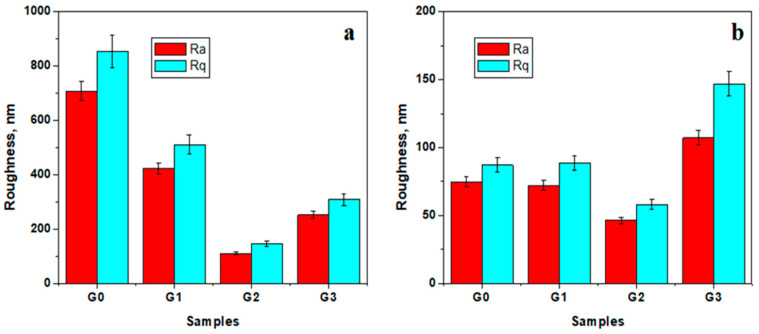
Surface roughness values measured at a scanning area of (**a**) 20 μm × 20 μm and (**b**) 5 μm × 5 μm.

**Table 1 biomedicines-12-00694-t001:** The composition of experimental hydrogels.

Hydrogels	Components wt%	P1 *	P2 *	Nano Capsules with P1	pH
G0	PEG 400—28	-	-	-	6
G1	Fumed silica—8	0.5%	-	-	7
G2	Distilled water—58	-	0.5%	-	6
G3	Hydroxyapatite—6	-	-	1%	6

P1 *—Anti-Amelogenin, X isoform, ABT260; P2 *—Anti-Kallikrein L1, K3014.

**Table 2 biomedicines-12-00694-t002:** Descriptive statistics for the viability values of the cytotoxicity test.

Materials	N Analysis	N Missing	Mean	Standard Deviation	SE of Mean
G0	3	0	89.32175	0.96683	0.5582
G1	3	0	90.96621	1.32606	0.7656
G2	3	0	88.68208	1.01523	0.58614
G3	3	0	89.18144	0.86029	0.49669

**Table 3 biomedicines-12-00694-t003:** One-way ANOVA test results of the investigated hydrogels.

	DF	Sum of Squares	Mean Square	F Value	Prob > F
Model	3	8.83885	2.94628	2.64004	0.12108
Error	8	8.92799	1.116		
Total	11	17.76685			

Null Hypothesis: The means of all levels are equal. Alternative Hypothesis: The means of one or more levels are different. At the 0.05 level, the population means are not significantly different. After performing the one-way ANOVE test, no significant differences were observed among the gels. The value of *p* = 0.12108.

## Data Availability

The original contributions presented in the study are included in the article, further inquiries can be directed to the corresponding author.

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
