# Peer review of "Developing Bioactive Hydrogels with Peptides for Dental Application"

_biomedicines, 2024, doi:10.3390/biomedicines12030694_

Round 1
Reviewer 1 Report
Comments and Suggestions for Authors
The authors developed a bioactive hydrogel containing Leucine Rich Amelogenin Peptide to apply for enamel remineralization. They characterized the physical-chemical properties of the hydrogel, as well as its antibacterial effects and cytotoxicity. However, the authors did not provide direct evidence of how the hydrogel promotes enamel remineralization. The application of LRAP in enamel remineralization is not new and has been investigated in several prior studies (e.g., Shafiei et al., Scanning, 2015; Mukherjee et al., J Mater Res, 2016 and Kwak et al., J Dent Res, 2017). Despite the authors' assertion in the concluding paragraph that the impact against S. mutans is an encouraging result, the mechanism underlying the peptides' antimicrobial effects remains unclear and requires further elucidation. In addition, despite the authors using a different hydrogel for peptide delivery compared to prior research (e.g., Mukherjee et al., J Mater Res, 2016), the rationale, novelty, and advantages of using this hydrogel were not clear.
Some specific points of concern include:
1. Line 24: The author stated that the study ‘assessed the characteristics of peptides in terms of enamel remineralization initiations,’ yet this aspect was not demonstrated in the analyses conducted in this study.
2. Line 25: The author mentioned that the analysis in this study ‘emphasizes the ability of peptides to promote cell adhesion.’ However, the cell experiment conducted in this study focused on cytotoxicity rather than assessing cell adhesion.
3. Line 200: The authors indicated that G1, G2, and G3 containing peptides exhibited an absorption peak at 219 nm. However, the prominence of the 219 nm peak in G2 is not evident in Figure 2. In addition, it is suggested that the legend in Figure 2 be organized in a sequential order.
4. Line 236: The nanostructure observed cannot be confirmed as nHAP or nSiO2 without further elemental or phase analysis.
5. The method of referencing figures is inappropriate; for example, in Line 237, Figure 4a should be enclosed within brackets at the end of the sentence. This issue was found throughout the manuscript and should be rectified accordingly.
6. In Line 286, the authors abruptly introduced chitosan without earlier context.
7. The authors used AFM to analyze the surface properties and toughness of hydrogels. However, the relevance of these properties to the hydrogel's function in enamel remineralization requires clarification. The rationale for conducting such analyses should be elucidated.
8. English editing is necessary for this manuscript.
Comments on the Quality of English LanguageImproving the grammar and careful editing and proof-reading in necessary.
Author Response
Dear Reviewer,
Sincere thanks for the careful review of the manuscript. We appreciate your efforts and feedback. We have utilized your observations to enhance the quality of our work. We are grateful for your valuable contribution to this process. Below, you will find our responses.
1. Line 24: The author stated that the study ‘assessed the characteristics of peptides in terms of enamel remineralization initiations,’ yet this aspect was not demonstrated in the analyses conducted in this study.
Response: It's true, but this paper represents the first part of the study in which we aimed to characterize the material to choose the optimal composition for the ultimate purpose, namely enamel remineralization.
2. Line 25: The author mentioned that the analysis in this study ‘emphasizes the ability of peptides to promote cell adhesion.’ However, the cell experiment conducted in this study focused on cytotoxicity rather than assessing cell adhesion.
Response: In this study, we conducted the initial cytotoxicity analyses to select the appropriate peptides for the intended purpose. We will further delve into the study of this peptide through comprehensive cytotoxicity studies and SEM. In this particular study, we did not have the opportunity to perform such a complex analysis.
3. Line 200: The authors indicated that G1, G2, and G3 containing peptides exhibited an absorption peak at 219 nm. However, the prominence of the 219 nm peak in G2 is not evident in Figure 2. In addition, it is suggested that the legend in Figure 2 be organized in a sequential order.
Response: We changed the legend of the Figure 2 and we correct in the text all you suggest us.
4. Line 236: The nanostructure observed cannot be confirmed as nHAP or nSiO2 without further elemental or phase analysis.
Response: The determination was carried out using an SEM apparatus that is not equipped with EDAX. The preparation of hydrogels was conducted within our laboratory, and for this study, we did not have the opportunity to perform this analysis.
5. The method of referencing figures is inappropriate; for example, in Line 237, Figure 4a should be enclosed within brackets at the end of the sentence. This issue was found throughout the manuscript and should be rectified accordingly.
Response: We changed.
6. In Line 286, the authors abruptly introduced chitosan without earlier context.
Response: We added in the text, line 98-100.
7. The authors used AFM to analyze the surface properties and toughness of hydrogels. However, the relevance of these properties to the hydrogel's function in enamel remineralization requires clarification. The rationale for conducting such analyses should be elucidated.
Response: Atomic Force Microscopy (AFM) is a powerful tool for the investigation of fine microstructural details and for exploring the nanostructural level of composite materials. Therefore, we employ AFM to characterize the experimental hydrogel fine microstructure (at scanned area of 20 μm x 20 μm) and their nanostructure (at a scanned area of 1 μm x 1 μm), please see Figure 5. The higher magnification and resolution of AFM investigation allows observing filler particles cross-linking with the gel matrix and the mineral nanoparticles distribution among organic matrix was observed and discussed. Tridimensional profiles evidence better the filler particles interaction with organic matrix and supports the surface roughness measuring (e.g. Ra and Rq parameters) which was measured and its variation was discussed.
8. English editing is necessary for this manuscript.
Response: Thank you for your feedback. We have already conducted English editing for the manuscript, and we believe it now meets the required standards.
Reviewer 2 Report
Comments and Suggestions for Authors
I appreciate the opportunity to review the manuscript on tooth decay, a prevalent issue, particularly among younger populations due to enamel demineralization. Despite progress in oral hygiene, billions are affected. Amelogenin, a key enamel protein, influences bio-mineralization. This in vitro study explores peptide-containing hydrogels, emphasizing their potential in enamel remineralization through SEM, AFM, UV-VIS, FTIR analyses. Peptides show promise in promoting cell adhesion and treating early carious lesions. In conclusion, short chain peptides in hydrogels offer potential for individual or professional dental care. However, the study is very interesting; I suggest providing some recommendations to improve the manuscript.
- 1. The methodologies did not mention the number of analyzed samples. I strongly recommend including the number of samples for each experiment in the Materials and Methods section as well as in the figures.
- 2. Figure 3, depicting antibacterial activity, could benefit from incorporating numerical values on the horizontal axis for clarity. Consider adding confidence intervals representations and indicating whether bacteria are sensitive or resistant to the hydrogels.
- 3. If the samples are n=3, did you perform a normal distribution to use parametric analysis? Non-parametric analysis might be more appropriate, e.g., in Table 2.
- 4. In the AFM methods, describe the acronyms Ra and Rq for better understanding.
- 5. The discussion section, particularly from line 336 to 382, includes theoretical and background information without substantial discussion. It is essential to discuss each section, such as FTIR, UV-VIS, etc., and their respective results.
- 6. Please include in the discussion the limitations of the study and provide some perspectives for the future.
Regards,
Author Response
Dear Reviewer,
We attached the document with our responses.

Reviewer 3 Report
Comments and Suggestions for Authors
The manuscript entitled "Developing bioactive hydrogels with peptides for dental application", from the authors Alexandrina Muntean, Codruta Sarosi, Ioan Petean, Stanca Cuc, Rahela Carpa, Ioana Andreea Chis, Aranka Ilea, Ada Gabriela Delean and Marioara Moldovan.
In general, the manuscript is good. Experiments are well planned, methods and tests are well chosen. Some corrections and explanations are also needed.
Hydrogels are not synthesized, but prepared by mixing components, (line 96).
Table 1 does not clearly show the composition of the samples. I ask the authors to state the amount of components in the composition of the samples: how much is in the composition of the hydrogel PEG 400, fumed silica, distilled water and hydroxyapatite?
It is unclear how the FTIR spectrum of the samples was recorded (FT-IR Spectroscopy, lines 95-100). Are the prepared hydrogels (which have water in them) applied in the form of a thin layer on the ATR device? It is not enough to simply state that the ATR technique was applied. Water will cover with its absorption the valence absorption region of the OH group, n(OH). I ask the authors to clarify.
The authors label the band at 1645 cm-1 as Amide band I (lines 178 and 179) and that is correct. How do the authors explain the absorption of sample G0 at 1645 cm-1 (Figures 1a and 1b) if it is a control sample without peptide? PEG 400, fumed silica and hydroxyapatite do not have amide groups. I ask the authors to clarify.
The text of the manuscript states: "In the spectrum of G1 and G2 hydrogels are presented the specific peak absorption of the peptides, and appear new peaks specific to hydroxyapatite, 570, 601 and 3571 cm−1" (lines 182 and 183). This cannot be seen in figures 1a and 1b for samples G1 and G2. No increased absorption is observed at 3571 cm-1, only a bend appears but at frequencies lower than 3500 cm-1. I ask the authors to clarify.
Figure 3 is a vague representation of antimicrobial activity. The labeling of the Y-axis with two labels is unclear. There is no explanation in the text either why two marks are used on the Y-axis? Also, the text of the manuscript states "This was the highest value obtained for these samples. In samples G1 (gel with P1) and G2 (gel with P2.) a low inhibition was recorded (Fig. 4)" (lines 220 - 222). Hence, the authors refer to figure 4 and figure 4 is "SEM images of the experimental hydro-gels general aspect: a) G0, b) G1, c) G2, d) G3 and fine microstructure: e) G0, f) G1 , g) G2 and h) G3" (lines 242 and 243). I ask the authors to correct the mistake.
In line 266 is not Figure 1g. I ask the authors to correct the mistake.
In lines 311 and 312 is not Figure 2h. I ask the authors to correct the mistake.
I consider that manuscript should be published in the journal "Biomedicines" after minor corrections.
Author Response
Dear Reviewer,
Sincere thanks for the careful review of the manuscript. We appreciate your efforts and feedback. We have utilized your observations to enhance the quality of our work. We are grateful for your valuable contribution to this process. Below, you will find our responses.
- Hydrogels are not synthesized, but prepared by mixing components, (line 96).
Response: We changed.
- Table 1 does not clearly show the composition of the samples. I ask the authors to state the amount of components in the composition of the samples: how much is in the composition of the hydrogel PEG 400, fumed silica, distilled water and hydroxyapatite?
Response: We added.
- It is unclear how the FTIR spectrum of the samples was recorded (FT-IR Spectroscopy, lines 95-100). Are the prepared hydrogels (which have water in them) applied in the form of a thin layer on the ATR device? It is not enough to simply state that the ATR technique was applied. Water will cover with its absorption the valence absorption region of the OH group, n(OH). I ask the authors to clarify.
Response: We have made changes, and we hope it is now clearer.
- The authors label the band at 1645 cm-1 as Amide band I (lines 178 and 179) and that is correct. How do the authors explain the absorption of sample G0 at 1645 cm-1 (Figures 1a and 1b) if it is a control sample without peptide? PEG 400, fumed silica and hydroxyapatite do not have amide groups. I ask the authors to clarify.
Response: We have made changes, and we hope it is now clearer.
- The text of the manuscript states: "In the spectrum of G1 and G2 hydrogels are presented the specific peak absorption of the peptides, and appear new peaks specific to hydroxyapatite, 570, 601 and 3571 cm−1" (lines 182 and 183). This cannot be seen in figures 1a and 1b for samples G1 and G2. No increased absorption is observed at 3571 cm-1, only a bend appears but at frequencies lower than 3500 cm-1. I ask the authors to clarify.
Response: We have made changes, and we hope it is now clearer.
- Figure 3 is a vague representation of antimicrobial activity. The labeling of the Y-axis with two labels is unclear. There is no explanation in the text either why two marks are used on the Y-axis? Also, the text of the manuscript states "This was the highest value obtained for these samples. In samples G1 (gel with P1) and G2 (gel with P2.) a low inhibition was recorded (Fig. 4)" (lines 220 - 222). Hence, the authors refer to figure 4 and figure 4 is "SEM images of the experimental hydro-gels general aspect: a) G0, b) G1, c) G2, d) G3 and fine microstructure: e) G0, f) G1 , g) G2 and h) G3" (lines 242 and 243). I ask the authors to correct the mistake.
Response: We have made changes, and we hope it is now clearer.
For figure 3, on the OY axis are represented the comparison of samples between them (G0 with G1; G0 with G2; G0 with G3, etc.), because this represents the Tukey statistical test. It is an automatically generated figure following the application of the One-Way ANOVA statistical test and then Tukey to obtain more accurate data regarding the existing differences between each group of investigated samples.
- In line 266 is not Figure 1g. I ask the authors to correct the mistake.
Response: We correct.
- In lines 311 and 312 is not Figure 2h. I ask the authors to correct the mistake.
Response: We correct.
Round 2
Reviewer 1 Report
Comments and Suggestions for Authors
My comments to 2nd revision is in red.
1. Line 24: The author stated that the study ‘assessed the characteristics of peptides in terms of enamel remineralization initiations,’ yet this aspect was not demonstrated in the analyses conducted in this study.
-This problem was not resolved.
In order to characterize and optimize material for the ultimate purpose of enamel remineralization, remineralization experiments need to be performed. If this optimization is out of the scope of this study, the authors need to justify as how all the experiments performed are relevant to enamel remineralization.
How peptide conformation in the hydrogel related to its potential ro remineralizer. What is the relevant of all the structures seen on AFM and SEM? These were applied on a solid surface (glass) and not enamel. The behavior of these gels can be totally different when a bioactive material like enamel is the substrate.
The sentence in line 16-28 emphasizes “in terms of enamel remineralization” . However, there is no explanation as why certain gels/peptide can be more effective for enamel remineralization.
The authors need to rewrite the aim of their study and provide justification for all the experiments.
Response: It's true, but this paper represents the first part of the study in which we aimed to characterize the material to choose the optimal composition for the ultimate purpose, namely enamel remineralization.
2. Line 25: The author mentioned that the analysis in this study ‘emphasizes the ability of peptides to promote cell adhesion.’ However, the cell experiment conducted in this study focused on cytotoxicity rather than assessing cell adhesion.
Response: In this study, we conducted the initial cytotoxicity analyses to select the appropriate peptides for the intended purpose. We will further delve into the study of this peptide through comprehensive cytotoxicity studies and SEM. In this particular study, we did not have the opportunity to perform such a complex analysis.
-There is a difference between cell adhesion and cytotoxicity. The later refers to a substrate (agent) being toxic to the cells. Cell adhesion refers to the ability of the cells to interact with the surface. The authors need to be consistent as what they are measuring and clarify it in the tex. What is measured in their experimental assay of “co-culturing”. Please clarify this in the tex.
3. Line 200: The authors indicated that G1, G2, and G3 containing peptides exhibited an absorption peak at 219 nm. However, the prominence of the 219 nm peak in G2 is not evident in Figure 2. In addition, it is suggested that the legend in Figure 2 be organized in a sequential order.
Response: We changed the legend of the Figure 2 and we correct in the text all you suggest us.
4. Line 236: The nanostructure observed cannot be confirmed as nHAP or nSiO2 without further elemental or phase analysis.
Response: The determination was carried out using an SEM apparatus that is not equipped with EDAX. The preparation of hydrogels was conducted within our laboratory, and for this study, we did not have the opportunity to perform this analysis.
5. The method of referencing figures is inappropriate; for example, in Line 237, Figure 4a should be enclosed within brackets at the end of the sentence. This issue was found throughout the manuscript and should be rectified accordingly.
Response: We changed.
6. In Line 286, the authors abruptly introduced chitosan without earlier context.
Response: We added in the text, line 98-100.
7. The authors used AFM to analyze the surface properties and toughness of hydrogels. However, the relevance of these properties to the hydrogel's function in enamel remineralization requires clarification. The rationale for conducting such analyses should be elucidated.
Response: Atomic Force Microscopy (AFM) is a powerful tool for the investigation of fine microstructural details and for exploring the nanostructural level of composite materials. Therefore, we employ AFM to characterize the experimental hydrogel fine microstructure (at scanned area of 20 μm x 20 μm) and their nanostructure (at a scanned area of 1 μm x 1 μm), please see Figure 5. The higher magnification and resolution of AFM investigation allows observing filler particles cross-linking with the gel matrix and the mineral nanoparticles distribution among organic matrix was observed and discussed. Tridimensional profiles evidence better the filler particles interaction with organic matrix and supports the surface roughness measuring (e.g. Ra and Rq parameters) which was measured and its variation was discussed.
-Please see my comment to concern #1 above. How the surface roughness is relevant to ability of an agent to remieralize enamel. What is an ideal microstructure to promote enamel remineralization? What is the function of these nanoparticles?
8. English editing is necessary for this manuscript.
Response: Thank you for your feedback. We have already conducted English editing for the manuscript, and we believe it now meets the required standards.
-The text was improved but there is still room for improvement.
The discussion is repetitive. References to previous studies using chitosan and LRAP are needed.
Discussion needs to be revised and focused on the findings of the present study and their relevance to enamel remineralization. The advantages of the hydrogel with compared to what has been already published (see some references below).
-Repairing human tooth enamel with leucine-rich amelogenin peptide-chitosan hydrogel (vol 31, pg 556, 2016)
K Mukherjee, et al 2016
JOURNAL OF MATERIALS RESEARCH 31 (6), 821-821
- Mukherjee K, et al 2021
Amelogenin Peptide-Chitosan Hydrogel for Biomimetic Enamel Regrowth. Front Dent
Med. 2021;2:697544. doi: 10.3389/fdmed.2021.697544. Epub 2021 Jun 16. PMID:
37900722; PMCID: PMC10611442.
-Mukherjee K, et al 2019
Peptide-Mediated Biomimetic Regrowthof Human Enamel In Situ. Methods Mol Biol. 2019;1922:129-138. doi:
10.1007/978-1-4939-9012-2_13. PMID: 30838571; PMCID: PMC8006743.
-Dogan et al 2018,
ACS Material Science and Engineering
2018, 4, 5, 1788–1796
-Ding et al 2020
Regenerative Biomaterials, Volume 7, Issue 3, June 2020, Pages 283–292,
Author Response
Dear reviewer,
We hope that this time the modifications we have made are in accordance with your requirements. We have added the recommended references and completed the discussion section. Attached is a document with all the responses (written in blue) to your questions.

Reviewer 2 Report
Comments and Suggestions for Authors
The manuscript was improved and addressed the suggestions
Author Response
Dear reviewer,
We are pleased that the modifications made are in accordance with your requirements. We appreciate your suggestions and have worked diligently to improve the manuscript accordingly. Your insights were invaluable, and we believe the revisions made enhance the overall quality of the manuscript.
Round 3
Reviewer 1 Report
Comments and Suggestions for Authors
The manuscript has significantely improved from its original submission. It is appropriate to be accepted.